# Importance of the Cysteine-Rich Domain of Snake Venom Prothrombin Activators: Insights Gained from Synthetic Neutralizing Antibodies

**DOI:** 10.3390/toxins16080361

**Published:** 2024-08-15

**Authors:** Laetitia E. Misson Mindrebo, Jeffrey T. Mindrebo, Quoc Tran, Mark C. Wilkinson, Jessica M. Smith, Megan Verma, Nicholas R. Casewell, Gabriel C. Lander, Joseph G. Jardine

**Affiliations:** 1Department of Immunology and Microbiology, Scripps Research Institute, La Jolla, CA 92037, USA; lmisson@scripps.edu (L.E.M.M.); qtran@scripps.edu (Q.T.);; 2IAVI Neutralizing Antibody Center, Scripps Research Institute, La Jolla, CA 92037, USA; 3International AIDS Vaccine Initiative, New York, NY 10004, USA; 4Department of Integrative Structural and Computational Biology, Scripps Research Institute, La Jolla, CA 92037, USA; jmindrebo@scripps.edu (J.T.M.); glander@scripps.edu (G.C.L.); 5Centre for Snakebite Research & Interventions, Department of Tropical Disease Biology, Liverpool School of Tropical Medicine, Liverpool L3 5QA, UK; mark.wilkinson@lstmed.ac.uk (M.C.W.); nicholas.casewell@lstmed.ac.uk (N.R.C.)

**Keywords:** snake venom metalloproteinases, prothrombin activator, *Echis*, recombinant antivenom, neutralizing antibody, cysteine-rich domain, ecarin structure

## Abstract

Snake venoms are cocktails of biologically active molecules that have evolved to immobilize prey, but can also induce a severe pathology in humans that are bitten. While animal-derived polyclonal antivenoms are the primary treatment for snakebites, they often have limitations in efficacy and can cause severe adverse side effects. Building on recent efforts to develop improved antivenoms, notably through monoclonal antibodies, requires a comprehensive understanding of venom toxins. Among these toxins, snake venom metalloproteinases (SVMPs) play a pivotal role, particularly in viper envenomation, causing tissue damage, hemorrhage and coagulation disruption. One of the current challenges in the development of neutralizing monoclonal antibodies against SVMPs is the large size of the protein and the lack of existing knowledge of neutralizing epitopes. Here, we screened a synthetic human antibody library to isolate monoclonal antibodies against an SVMP from saw-scaled viper (genus *Echis*) venom. Upon characterization, several antibodies were identified that effectively blocked SVMP-mediated prothrombin activation. Cryo-electron microscopy revealed the structural basis of antibody-mediated neutralization, pinpointing the non-catalytic cysteine-rich domain of SVMPs as a crucial target. These findings emphasize the importance of understanding the molecular mechanisms of SVMPs to counter their toxic effects, thus advancing the development of more effective antivenoms.

## 1. Introduction

The World Health Organization (WHO) estimates that 5.4 million people worldwide are bitten by snakes annually. Between a third to a half of these snakebites will result in envenoming, leading to the death of at least 138,000 people [1]. Those numbers are likely erroneous, as under-reporting of snakebite incidence and mortality is common in the affected areas and populations (mostly agricultural workers and children in low- or middle-income countries). Consequently, in 2017, the WHO formally listed snakebite envenoming as a highest-priority neglected tropical disease [2]. Snake venoms are complex mixtures of biologically active molecules that have evolved to immobilize prey, and are responsible for causing severe pathology and toxicity following envenoming in humans [1]. The only effective treatment for snakebite is the use of antivenoms, i.e., antibody therapies that neutralize the effects of snake venom toxins [3]. However, the manufacturing process of antivenoms has remained largely unchanged since the late 19th century, hyperimmunized animals being the source of polyclonal antibody mixtures injected into patients [4]. While antivenoms have saved countless lives, their side effects can be severe, and their efficacy remains relatively low [5]. In recent years, efforts have emerged to develop better antivenoms, especially with the use of monoclonal antibodies [6,7,8]. The combination of monoclonal antibodies or nanobodies into oligoclonal mixtures enables the neutralization of multiple toxins with a single cocktail [9,10]. Therefore, tailored control of the antivenom composition is essential, requiring an in-depth understanding of which toxins are present in the venom, their effects and their activity. Incorporating these factors into antivenom development affords us the capacity to specifically target key toxins, or entire classes of toxins, to reduce the whole effects of envenoming [11].

Numerous proteins contribute to snake venom toxicity [12,13]. Among these, snake venom metalloproteinases (SVMPs), phospholipases A2 (PLA_2_s), snake venom serine proteases (SVSPs) and three-finger toxins (3FTxs) are the major contributors to morbidity and mortality, and neutralizing these specific toxin families is crucial for mitigating the harmful effects of envenoming [14]. Previous studies have shown that neutralizing a single toxin component can substantially counteract the pathological effects of the whole venom. Additionally, synergies among toxins have been documented [15,16], suggesting that inhibition of a single toxin family could alleviate envenoming effects by disrupting these interactions. For instance, repurposing the PLA_2_ inhibitor varespladib has shown promise in preventing local tissue damage caused by the combined activity of PLA_2_s and cytotoxic 3FTxs in spitting cobra venom [17]. Recently, several groups have shown that neutralizing antibodies against long-chain 3FTx can protect mice from lethal venom challenges [18,19]. It has also been shown that inhibiting SVMP activity can prevent viperid venom-induced hemorrhage and dermonecrosis [20,21,22,23].

Snake venom proteases (metalloproteinases and serine proteases) are one of the major components responsible for the toxicity of viper venom [24,25,26]. Their effects range from causing tissue damage to hemorrhage and disruption of the blood clotting system, which can lead to fatal consequences in envenomated victims [27]. The primary pathological effect of SVMP is disruption of hemostasis, either via hemorrhage [26,28], which occurs through degradation of vascular endothelium components [29], and/or by targeting various factors in the blood coagulation cascade [30]. Both venom gland transcriptomics and venom proteomics indicate that snake venoms from the *Echis* genus, commonly referred to as saw-scaled vipers, contain high levels of SVMPs, although disparities between toxin gene transcription and protein abundance in the venom have been observed [31]. Proteomic analysis, contingent upon various factors such as the selected database (for example, *Viperidae* taxid 8689, *Echis* taxid 8699 or *Echis carinatus* taxid 40353), species (*E. carinatus carinatus* [32,33], *E. carinatus sochureki* [31], *E. ocellatus* [31,34], *E. coloratus* [31], *E. pyramidum leakeyi* [31]) and the chosen quantification methods, reveals SVMP fractions ranging from 5 to 70% of the whole venom [35]. Importantly, the inactivation of SVMPs from *Echis* vipers by small molecules such as zinc chelators or peptide inhibitors has been demonstrated to mitigate both local and systemic toxicity [23,36,37]. Therefore, developing antibodies that specifically target SVMPs could prove efficacious against envenomation from snakes with high concentration of SVMPs in their venom.

SVMPs are categorized into class I (SVMP PI), II (SVMP PII) or III (SVMP PIII). SVMP PIs comprise solely a zinc-dependent metalloproteinase domain (M) required for proteolytic activity. SVMP PIIs feature an additional disintegrin domain (D), while SVMP PIIIs contain both M and D-like domains, along with a C-terminal cysteine-rich domain (C) [38]. Notably, the C domain is believed to play a critical role in recruiting specific substrates in both SVMPs and the related ADAM class of human enzymes [25]. SVMP PIIIs that affect the blood coagulation cascade usually activate either of two key factors—Factor X and II (prothrombin)—thereby exhibiting procoagulant activity [30]. SVMP PIIIs that operate on Factor II are referred to as snake venom prothrombin (PT) activators and are categorized into two groups based on their cofactor requirements. Group A and B comprise Ca^2+^-independent and Ca^2+^-dependent metalloproteinases, respectively [39]. The most extensively studied example within group A is ecarin, identified in ‘*Echis carinatus*’ venom in 1975 [40]. It is worth mentioning that often at that time, the origin region of the snake was unspecified, as the sources frequently included snakes from zoos [41]. Therefore, it is challenging to determine the specific snake species used for the initial studies on ecarin, particularly since the taxon ‘*Echis carinatus*’ has since been reclassified into several species. As a result, the name ecarin has been commonly used to describe group A snake venom SVMP PIIIs activating PT from the *Echis* genus.

In this study, we describe the isolation and characterization of antibodies against two recombinant SVMP PIII PT activators from *Echis pyramidum leakeyi* (ecarin) and *Echis romani* (EoMP06) using a synthetic Fab library displayed on yeast. We employed a library screening strategy to identify cross-reactive antibodies that bind to both target SVMPs. Following recombinant production and characterization, we identified three clones with strong neutralizing activity against both SVMPs. Importantly, we demonstrated the biological relevance of our neutralizing antibodies by confirming their activity against the native PT activators from three African saw-scaled vipers: *E. p. leakeyi*, *E. romani* and *E. leucogaster*. Structural analysis of one neutralizing antibody in complex with recombinant ecarin revealed the non-catalytic C domain as the neutralizing epitope. Despite their lack of enzymatic activity, the C domains of SVMP PIIIs can significantly contribute to SVMP toxic effects through protein–protein interactions. These interactions may include binding to platelets [42,43], to factors within the blood coagulation cascade [44] or to various substrates like collagens [45,46] or α1β1 integrins [47]. These findings suggest that our antibodies likely block PT activity by disrupting SVMP substrate targeting. Overall, our study highlights the critical role of the C domain of snake venom PT activators in conferring toxicity and identifies it as a promising epitope for the development of neutralizing monoclonal antibodies.

## 2. Results

### 2.1. Recombinant Ecarin (rEcarin) and EoMP06 (rEoMP06) Expression and Activity

Ecarin is an SVMP PIII that activates PT in a Ca^2+^-independent manner [39] by cleaving the Arg320-Ile321 peptide bond, forming the active meizothrombin intermediate (Appendix A) [48]. A recombinant form of ecarin (rEcarin) from Kenyan *E. p. leakeyi* venom [49] (formerly classified as *Echis carinatus*) was produced by synthesizing the pro-, M, D and C domains (Ile20-Tyr616 from Uniprot sequence Q90495) into a mammalian expression vector containing a secretion signal and a His_10_-tag. Ecarin contains a prodomain (Ile20-Ile189) essential for proper folding; however, hydrolysis of this prodomain is necessary for the enzyme to become active [50]. As the prodomain is partially cleaved during protein expression [51], no additional steps were needed to characterize its activity (Appendix A).

EoMP06, another SVMP PIII PT activator, shares 90.6% sequence identity with ecarin over the M, D and C domains (Val90-Tyr515). The sequence for EoMP06 (Uniprot sequence Q6X1T6) was originally obtained from the cDNA of the venom gland of the West African viper *Echis ocellatus* [52] (since reclassified as *Echis romani* [53,54]). To express the recombinant EoMP06 (rEoMP06), the coding sequence was synthesized in the same expression vector as rEcarin; however, the initial expression attempts did not produce any protein. Sequence alignments of ecarin and EoMP06 suggested that the EoMP06 prodomain from Uniprot may be incomplete (Appendix A), which we hypothesized could account for its unsuccessful expression. To address this, we inserted the missing prodomain fragment, consisting of residues Ile20 to Ser81 from the ecarin sequence, into the EoMP06 sequence, enabling successful expression of a His_10_-tagged rEoMP06. Similar to rEcarin, the prodomain of rEoMP06 was partially cleaved during protein expression (Appendix A).

Recombinantly expressed human PT (Uniprot sequence P00734) was used as a substrate for both rEcarin and rEoMP06. PT activation was monitored via the formation of para-nitroaniline (pNA) at 405 nm from S-2238, a chromogenic substrate for meizothrombin and thrombin. Both rSVMPs exhibit similar activity, and no significant activity could be detected in the presence of 20 mM EDTA, which chelates the catalytic zinc (Appendix A). These findings demonstrate that our recombinantly expressed SVMPs are natively folded and active.

### 2.2. Isolation of rEcarin Antibodies from a Synthetic Human Fab Library

Antibodies against rEcarin were isolated from a synthetic human antibody library, which contains 6 × 10^10^ unique antibodies displayed as fragment antigen-binding (Fabs) on the surface of *Saccharomyces cerevisiae*. The library diversity was introduced into the complementarity-determining region 3 of the heavy chain (CDRH3), with CDRH3 loop lengths ranging from 10 to 20 residues [18]. The rEcarin-binding clones were enriched using a combination of magnetic-activated cell sorting (MACS) followed by fluorescence-activated cell sorting (FACS) (Figure 1A). Selections alternated between positive-affinity sorts (AFFs) to enrich rEcarin binders and negative polyreactive reagent sorts (PSRs) to deplete non-specific binders [18,55]. After each selection, the collected cells were expanded and induced for further rounds of selection. A subset was also set aside for deep sequencing analysis. After the MACS/FACS selections with rEcarin, a final selection was performed using rEoMP06 to identify the antibodies that recognized a conserved epitope between the two SVMPs (X-AFF).

Following the selections, the antibody-encoding DNA was harvested from the cells reserved after the expansions and deep sequence analyzed. Based on these data, 96 antibodies were selected for reformatting as IgG and subsequent expression, with priority given to clones that bound to both rEcarin and rEoMP06. Of these, 72 antibodies bound to rEcarin by enzyme-linked immunosorbent assay (ELISA) and exhibited minimal or no off-target binding to single-strain DNA (ssDNA) or detergent-solubilized CHO cell membrane preparations (CHO-SMPs) (Figure 1B).

### 2.3. Identification of rEcarin and rEoMP06 Neutralizing Antibodies

We next screened the 72 antibodies in our rEcarin activity assay to identify those with function-blocking activity. The assay was set up as described in the Materials and Methods (Section 5.6. SVMP rate calculation) but with the addition of a single high concentration (0.1 mg/mL) of antibody. After 30 min, antibodies B3, H11 and H12 showed inhibition comparable to the negative control (20 mM EDTA) (Figure 1C).

We further evaluated the three antibodies at different antibody-to-rSVMP molar ratios (rEcarin or rEoMP06) and at different timepoints to determine the rates of reaction. To ensure that the S-2238 concentration would not affect the reaction rates, only pNA concentrations below 50 µM were plotted (representing 10% substrate consumption) (Appendix A). For both rSVMPs, H11 and H12 exhibit stronger inhibition against rEcarin compared to B3 (Figure 2A). Notably, the three antibodies are from the V_H_1-02 class of antibodies, and their sequences are highly similar and feature multiple negatively charged residues, suggesting they are recognizing a common epitope (Figure 2B). To test this hypothesis, a biolayer interferometry-based (BLI) competition assay was used, in which the three antibodies competed against each other for ecarin binding, which confirmed their binding to a common epitope (Figure 2C). Subsequently, we selected H11 for further characterization due to its superior binding kinetics (Figure 2D). Specifically, H11 has an estimated *K*_D_ of 87 ± 8 nM, whereas B3 and H12 showed *K*_D_ of 489 ± 38 nM and 149 ± 12 nM, respectively (Appendix A).

We then sought to evaluate the effects of H11 on either substrate binding or on the ecarin catalytic constant using Michaelis–Menten kinetics and determine the type of inhibition. As previously described, the rEcarin affinity for PT is low [51]. In our experimental setup, rEcarin could not be saturated with PT and we were only able to estimate a *K*_M_ value of 4 µM (Figure 3 and Appendix A). This limitation made further kinetic characterization challenging.

### 2.4. H11 Neutralizes Native Ecarin (nEcarin) In-House Purified from African Echis Venoms

Until this point, all work had been carried out using rEcarin and rEoMP06 expressed from a mammalian cell line, and we wanted to confirm that the neutralizing antibodies performed similarly against native SVMPs. Therefore, we performed the same single-antibody-concentration PT activation assay described above using commercially available purified ecarin from *Echis carinatus* (Sigma E0504). Surprisingly, none of the three previously identified rEcarin neutralizing antibodies showed any neutralizing activity on the purchased enzyme (Appendix A). SDS-PAGE gel electrophoresis revealed that the commercially available purified ecarin had an apparent molecular weight of 10–15 kDa (Appendix A). The calculated molecular weight of ecarin after processing the prodomain (Val190-Tyr615) is 51 kDa, and it contains five putative N-linked glycosylation sites (https://services.healthtech.dtu.dk/services/NetNGlyc-1.0/ (accessed on 10 May 2024)), [51] which would be expected to result in a molecular weight of 70–75 kDa. The smallest SVMPs, SVMP PIs, have a calculated molecular weight of 20–25 kDa. Despite exhibiting similar PT activation to rEcarin, this appears to be a fundamentally different protease and we decided not to further characterize the commercial preparation.

We then opted to test SVMP PIII PT activators from whole *Echis* venoms. Venoms were sourced from a total of four snakes, *E. p. leakeyi* (originating from Kenya), *E. romani* (originating from Nigeria, which before 2018 was classified as *E. ocellatus* [53,54]), *E. carinatus* (originating from India) and *E. leucogaster* (originating from Mali). H11 was tested for binding to 0.2 mg/mL whole venom by BLI, and reactivity was observed against the three African *Echis* venoms but not the Indian venom (Figure 4A).

Upon establishing H11 binding to native SVMP PIIIs, we next wanted to determine whether it could similarly neutralize the native proteins. To do this, SEC purification was used to separate out the ~50 kDa SVMP PIIIs from the SVMP PI and PII, and thrombin-like serine proteases (Figure 4B) that also cleave the S-2238 substrate and thus produce a positive signal (Appendix A) [32]. We isolated one SVMP PIII from *E. leucogaster* venom, which binds H11, and two SVMP PIIIs from *E. p. leakeyi* venom, one of which also binds H11 (Appendix A). The SVMP PIII fraction from the *E. romani* venom is more complex than either *E. leucogaster* or *E. p. leakeyi*, likely containing multiple SVMP PIIIs, which could not be resolved with a single SEC step (Appendix A). Additionally, we observed rapid precipitation of the *E. romani* SVMP PIIIs following SEC. Consequently, we did not further characterize the isolated *romani* SVMP PIIIs due to their instability. Instead, we investigated the potential effect of H11 on the two native PT activators from *E. leucogaster* and *E. p. leakeyi* venoms that bind our antibody.

We determined the rates of PT activation catalyzed by 50 nM of the SVMP PIII fractions of *E. leucogaster* and *E. p. leakeyi*. The rates in the presence of two concentrations of H11 (10 and 100 µg/mL) were compared to the rate of PT activation in the absence of H11 (Figure 4C). Although we could not directly compare the inhibition levels between native and recombinant ecarins, as we could only estimate the nEcarin concentrations, we were able to demonstrate that H11 effectively inhibits native PT activators from *E. leucogaster* and *E. p. leakeyi*.

### 2.5. Structural Characterization of the H11–Ecarin Complex

Having identified three competing neutralizing antibodies with similar CDRH3 motifs, we aimed to identify their epitope to define the mechanism of neutralization. To accomplish this, we solved a high-resolution structure of H11 Fab (Appendix A) in complex with rEcarin by cryogenic electron microscopy (cryo-EM). Initially, the H11–rEcarin complex was generated by incubating the two proteins at a 1:1 molar ratio for 15 min before sample vitrification. Data collection and processing using UltrAuFoil 1.2/1.3 holey gold grids yielded high-resolution 2D class averages, but only representing a limited number of views, which prohibited reconstructions with a directionally isotropic resolution (Appendix A).

To overcome issues with the preferred orientation, we prepared additional cryo-EM samples using holey gold grids coated with a graphene support film [56], which increased the diversity of particle orientations. Combining the graphene-grid dataset with our first dataset provided sufficient angular sampling to achieve a reconstruction of the H11 Fab–ecarin complex with a reported global resolution of ~3.4 Å. The final reconstruction reveals that H11 associates with the non-catalytic C domain adjacent to the M domain (Figure 5A,B). Importantly, the C domain is thought to play a crucial role in substrate recruitment and specificity [42,44,45,46,47,57], suggesting that H11 inhibits essential protein–protein interactions necessary for PT binding and processing.

The H11–rEcarin interface buries a total surface area of 688.4 Å^2^, with the V_H_ domain contributing 556.8 Å^2^ and the V_L_ domain providing 131.6 Å^2^. Despite our antibody library including fixed CDRH1 and CDRH2 regions, they constitute a significant portion of the interface. The two loops clamp around either side of helix 8 (residues 526–534) of the C domain, stabilizing the interaction through predominantly polar contacts (Figure 5C). The positioning of the CDRH1 and CDRH2 loops provides a scaffold to stabilize the highly acidic CDRH3 loop, which forms the central interaction network of the interface. Three aspartate residues in the CDRH3 loop—Asp97, Asp98 and Asp99—form the periphery of the interface by engaging in polar and charged interactions with the C domain and the H11 V_L_ domain. Notably, Asp97 forms a polar interaction network with Tyr32 from H11 CDRH1, as well as Gln535 and Tyr539 of the C domain, while Asp98 and Glu50 from CDRL2 interact electrostatically with Lys573 of the C domain (Figure 5D). Asp99 is positioned to form a tight hydrogen bond with Asn575, which in turn likely forms a lone-pair–π bonding interaction with Tyr34 of CDRL1 (Figure 5D).

At the central core of the interface, the remaining charged residues in the CDRH3 loop, Glu95 and Asp100, form a charged interaction network with Arg532 and Arg577, buried within a hydrophobic pocket (Figure 5E). Importantly, Glu95 is conserved across all three of our neutralizing antibodies (Figure 2B), underscoring the significance of this buried salt bridge for complex formation. However, Asp100 is present only in H11 and H12, which exhibit more favorable binding kinetics and slower off-rates than B3 (Figure 2D). Therefore, the addition of a second negatively charged residue to interact with Arg532 and Arg577 is likely critical for enhanced affinity and potency.

In addition to H11 blocking the putative substrate recruitment domain, we also identified a well-ordered density for a peptide bound to the active site of the M domain (Figure 5F). We modeled a seven-residue polyalanine peptide in this density, positioning the fourth bond adjacent to the catalytic zinc ion. These results were unexpected, as our cryo-EM samples were prepared with wild-type ecarin, which should retain proteolytic activity. However, the continuity of the peptide density suggested it had not undergone proteolytic cleavage, unless it represented a stably coordinated catalytic intermediate. Prodomain peptides could potentially have been present in our ecarin preparation [50,58,59], and might have been inadvertently trapped in our ecarin–H11 complex. Since we were unable to assign a sequence to this peptide, we propose that our structure represents a ternary complex of H11, ecarin, and an unknown peptide that cannot undergo hydrolysis. These findings suggest that H11 traps ecarin in a conformation mimicking the PT–ecarin complex, where PT is bound to the C domain and its respective cleavage site is positioned in the proteolytic active site for processing.

Ecarin belongs to the subclass SVMP PIII-a [38], which does not undergo any post-translational modifications (such as cleavage between the M and D/C domains for the subclass PIII-b, dimerization for the subclass PIII-c, or C domain complexation with lectin-like domains for the subclass PIII-d) besides prodomain removal. A comparison between ecarin from our model and previously solved crystal structures of SVMP PIIIs-a/b, (Vap2 [60], AaHIV [61], atragin [62], bothropasin [63]) shows that the D and C domains adopt a unique conformation in our structure. The long arm of the D domain rotates inwards, facilitating the formation of a small, polar interface between the C domain and the M domain (Figure 5G). However, all monomeric SVMP PIIIs-a/b adopt conformations where the C domain is rotated away from the M domain, forming few if any contacts (Appendix A). When comparing RVV-X [64], an SVMP PIII-d, to ecarin, both SVMPs exhibit a similar conformation across their three domains (Appendix A). Notably, RVV-X and ecarin show a higher sequence identity (64.5%) compared to ecarin with Vap2 (57.2%), AaHIV (54.5%), atragin (51.8%) or bothropasin (57.1%) for the M, D and C domains. This higher sequence identity may explain their structural similarity.

The C domain is considered the putative substrate recruitment domain for SVMP PIIIs [44,45,46,47,57,65], but to our knowledge, there are no structural studies that directly confirm these hypotheses. Therefore, we used AlphaFold 2 to generate a PT–ecarin complex to determine if H11 would inhibit PT association. To generate the model, we removed the first 189 residues of ecarin, corresponding to the autoinhibitory prodomain, and folded it with full-length PT lacking its signal and propeptide sequences (residues 1–43). AlphaFold generated five models that were all modestly well-scored, with the highest having a combined pTM and ipTM score of 0.755 (Figure 6A and Appendix A).

All five models formed a similar complex, and interestingly, positioned a long loop containing ecarin’s secondary cleavage site, Arg271-Thr272 [51], and not Arg320-Ile321, in the proteolytic active site (Figure 6A and Appendix A). Notably, the predicted model placed PT’s Kringle 2 domain (residues 213–291) in association with the C domain of ecarin (Figure 6A). Analysis of the electrostatic potential surfaces indicated that PT presented a predominantly negatively charged face to interact with the positively charged patches on the C and M domains (Figure 6B). While the confidence scores representing interface accuracy were too low for us to be sure of the exact placement of PT at the interface (Appendix A), the position of H11 at the C domain in our cryo-EM structure would block association with the 72 kDa PT molecule, regardless of substantial deviations from the predicted complex (Figure 6C). These results support a model where H11 sterically hinders a PT association by mimicking the ecarin–PT complex.

## 3. Discussion

Viper envenoming often leads to consumption coagulopathy, characterized by the activation and subsequent depletion of clotting factors [66]. This procoagulant property in *Echis* viper venoms is primarily due to potent PT activators [22], highlighting the importance of inhibiting these activators to mitigate the associated pathology. Our study showcases the utility of using recombinant monoclonal antibodies to neutralize the PT activity of SVMP PIIIs. Importantly, we also show that recombinantly produced SVMPs are viable surrogates for use in the selection and characterization process. Utilizing recombinant proteins allows for the production of highly pure preparations at scale [67]. It also enables the production of toxin variants, including domain truncations, the incorporation of epitope tags to aid in selections, and modifications to reduce proteolytic activity. Additionally, it ensures access to well-validated preparations, as the nomenclature in the field is continuously evolving and commercially available products are often poorly validated.

Overall, the presence of multiple domains in SVMPs often correlates with increased toxicity. SVMP PIIIs exhibit more potent hemorrhagic activity compared to SVMP PIs, which solely comprise the catalytic domain [26]. While targeting the M domain with peptide mimetics or zinc chelating compounds can effectively prevent venom-induced pathologies [20,21,22,23], neutralization of both the D and C domains has demonstrated the ability to mitigate local hemorrhage both in vitro and in vivo [68]. In addition, polyclonal antibodies neutralizing the D domain of SVMP PIIs and SVMP PIIIs from different rattlesnakes were shown to neutralize both proteolytic and hemorrhagic in vitro activities from crude *Crotalus atrox* venom [69,70]. Those studies underscore the potential of targeting the non-catalytic domains of SVMPs as a viable approach for the development of antivenoms. The three neutralizing antibodies identified in this campaign (B3, H11 and H12) all target the same epitope on ecarin. Conceptually, antibodies, similar to canonical inhibitors, can inhibit protease activity by binding at or in close proximity to the active site. Additionally, their larger size compared to conventional inhibitors allows them to block substrate access without necessarily requiring long insertion loops to the active site [71]. However, our cryo-EM structure analysis of H11 Fab in complex with ecarin revealed its binding away from the active site, instead targeting the C domain. This marks the first direct identification of a neutralizing epitope on the C domain of an SVMP PIII. Although we cannot exclude allosteric effects influencing ecarin activity upon H11 binding [72] due to a lack of kinetic data, our structural data, coupled with models of the ecarin–PT complex, strongly support a competitive mode of inhibition, although outside of the active site.

Despite the effectiveness of our antibodies against several African *Echis* PT activators, their efficacy did not extend to Indian *Echis* species. Several studies have pointed out potential differences in sequence and/or structure between PT activators found in African and Indian *Echis* venom. For instance, despite observing PT activation activity for the whole venom and the SVMP PIII fractions of *E. carinatus carinatus* (originated from India), the sequence of the Kenyan ecarin (Uniprot Q90495) could not be identified in this venom [32]. Another study highlights differences in immune response, where mice immunized with a recombinant D/C didomain of EoMP06 showed a good serum response for different African *Echis* venoms, but a weaker response for *Echis* venoms from Iran and Pakistan [68]. This suggests significant sequence differences between the PT activators of African and Indian *Echis* species. Moreover, our antibodies would not be effective against SVMP PIs or SVMP PIIs, as they target the C domain only present in SVMP PIIIs. This narrow range of efficacy highlights a major challenge in the development of a universal snake antivenom. To address this, it is important to include a panel of diverse toxins in the discovery process, incorporating both native toxins to capture the variety of native SVMPs, and recombinant SVMPs for biochemical characterization.

## 4. Conclusions

In conclusion, our study highlights the utility of combining recombinant and native PT activators to identify neutralizing epitopes, emphasizing the relevance of non-catalytic domains as crucial targets. We have elucidated the first structure of an SVMP PIII-neutralizing antibody complex, providing valuable insights into the mode of action of snake venom PT activators. These findings guide the generation of neutralizing antibodies against *Echis* PT activators and provide a framework for developing broadly neutralizing antibodies against various SVMP PIIIs. This approach has the potential to improve the efficacy and specificity of antivenoms, contributing to better therapeutic interventions for snakebite envenoming.

## 5. Materials and Methods

### 5.1. Magnetic-Activated Cell Sorting (MACS) of the Naïve Fab Library

MACS is used to deplete the Fab library of non-binding clones. Frozen aliquots of the naive Fab library [18] were thawed, and approximately 4 × 10^11^ cells were expanded in 18 L of synthetic-defined yeast medium without uracil (SD-Ura, Sunrise Science, Knoxville, TN, USA) and grown at 30 °C overnight. The following day, approximately 5 × 10^11^ cells were passaged into 18 L of SD-Ura and grown at 30 °C overnight. Cells were then pelleted and resuspended in 18 L of induction medium (20 g/L galactose, 1 g/L glucose, 6.7 g/L yeast nitrogen base, 5 g/L Bacto Casamino Acids, 38 mM disodium phosphate, 72 mM monosodium phosphate and 419 μM L-Trp), and approximately 4.5 × 10^11^ cells were induced at 18 °C for 3 days. The three MACS selections were performed on an autoMACS Pro Separator using autoMACS columns (Miltenyi Biotec, Bergisch Gladbach, Germany). For the first round of MACS (MACS1, Posseld2 program), 3 × 10^10^ induced yeast cells from our Fab naïve library were incubated with 50 nM biotinylated rEcarin for 1 h at 4 °C in PBS and 2% (*w*/*v*) bovine serum albumin (PBSA 2%). Cells were subsequently washed with PBSA 2% and incubated with 0.6 mL of Super Mag Streptavidin Beads (50 nm diameter Ocean Nanotech, San Diego, CA, USA) for 30 min at 4 °C in PBSA 2%. This step was repeated seven more times to sort a total of 2.4 × 10^11^ induced yeast cells (oversampling our Fab library diversity of 6.10^10^ by a factor of 4). The incubation time with rEcarin was limited to 1 h to prevent any proteolytic activity on the yeast surface display components. The cells binding to the magnetic beads were grown in SD-Ura o/n at 30 °C, and the following day induced o/n at 30 °C in induction medium. The second round of MACS (MACS2, Possel program) is a negative selection that only uses magnetic beads to remove Fab clones that bind to magnetic beads rather than to ecarin. Here, 2 × 10^10^ induced cells were incubated with 200 µL Super Mag Streptavidin Beads in PBSA 2% for 30 min at 4 °C. The non-binding cells were selected and immediately incubated with 50 nM rEcarin for 1 h 4 °C as described for MACS1. For MACS3 (Posseld2 program), cells binding to the magnetic beads were selected and grown in SD-Ura o/n at 30 °C.

### 5.2. Fluorescence-Activated Cell Sorting (FACS)

FACS is used to enrich the yeast cells in rEcarin-specific clones. To minimize the selection of polyspecific antibodies [73], we alternated between positive (affinity sorts AFF) and negative selections (polyspecific reagents or PSR sorts). After each round of selection, the sorted cells were expanded and re-induced prior to the next selection. In each round of selection, 1 to 5 × 10^7^ induced yeast cells were incubated for 60 min at 4 °C (rotating at 50 rpm) either with biotinylated rEcarin for AFF sorts, or biotinylated HEK-cell soluble membrane protein extracts [74] for PSR sorts in 500 µL PBSA 1% or PBS, respectively. To remove the excess of biotinylated antigen, yeast cells were then washed twice either with PBSA 1% for AFF sorts or PBS for PSR sorts and conjugated to 3 fluorophores (1µg/mL) for 20 min at 4 °C: streptavidin–allophycocyanin (APC) conjugate (Thermo Fisher Scientific, Waltham, MA, USA, cat #SA1005) was used to check rEcarin binding, anti-V5 antibody conjugated with Alexa Fluor 405 (AF405) to check the display of the heavy chain of the Fabs, and anti-C-Myc antibody conjugated with fluorescein isothiocyanate (FITC, Immunology Consultants Laboratory, Portland, OR, USA, cat #CMYC-45F) to check heavy-chain/light-chain parings of our Fabs. Yeast cells were then washed once and resuspended in PBSA 1% for sorting on a FACS Melody (BD Biosciences, Franklin Lakes, NJ, USA). Paired Fab chains were gated from a plot of AF405 versus FITC signal (heavy chain versus light chain), which was subsequently sorted for ecarin binding on a plot of APC versus FITC (paired heavy chain/light chain) [73]. Selected yeast cells were grown in SD-Ura medium and induced for the next selection rounds. For AFF1, AFF2 and AFF3, 100, 20 and 4 nM of biotinylated rEcarin were used, respectively. For cross-reactive sorting with rEoMP06, the AFF3 population was incubated with 200 nM of biotinylated rEoMP06. For PSR1 and PSR2, cells were incubated with 0.1 mg/mL of biotinylated HEK-cell soluble membrane protein extracts. The yeast cells from each sort were prepared for NGS sequencing.

### 5.3. Deep Sequencing Analysis

Cells were prepared for deep sequencing analysis as previously described [18]. Briefly, the plasmids of each yeast cell population were extracted, and the amplicons were subsequently amplified and indexed (xGen UDI primers (Integrated DNA Technologies, Coralville, IA, USA)) using two rounds of PCR. The amplicons were loaded on a MySeq System (Illumina) with a V3 600-cycle kit. Sequence quality of paired-end FASTQ files was analyzed using FastQC (v0.11.9) [75]. BBMerge (v38.87) [76] was used to merge forward and reverse reads, while VSEARCH (v2.15.1) [77] clustered merged reads to quantify identical sequences. Clustering was performed using the “cluster_fast” method to obtain FASTA files with sequence abundances. A Python custom script (Python 3.7) was used to parse clustered FASTA files, remove primers, identify the heavy chain and light chain and translate DNA to amino acids. Unique CDRH3 sequences were then counted for each heavy chain and light chain pair.

### 5.4. Gibson Cloning to Obtain Full Prodomain of EoMP06

The rSVMP base expression vector was digested with BsaI-HF V2 (NEB) restriction enzyme. The missing prodomain fragment to be inserted in the EoMP06 sequence was amplified by PCR from the ecarin sequence (Ile20-Ser81) with the following primers: 5′- GCTGCTGCCACAGGTGCCCATTCTATCATCCTGGGCAGCGG-3′ and 5′- GTGGTAGTAGCAATGATCTTCCACGCTAGGGTTTGTGGTGATCTCTC-3′ to generate a 300 bp fragment. The EoMP06 fragment was amplified by PCR with the following primers: 5′- GAGAGATCACCACAAACCCTAGCGTGGAAGATCATTGCTACTACCAC-3′ and 5′- ATGGTGATGATGGTGTCCGCTTCCGTAGGCGGTGTTCACGTCC-3′ to generate a 1600 kb fragment. We incubated 0.028 pmol of digested vector and 0.16 pmol of each insert at 50 °C for 1 h with the Gibson Assembly master mix (NEB). Then, we transformed 5 µL of the assembly reaction in NEB 5-alpha competent *E. coli* cells. Insertion of the missing prodomain was confirmed by Sanger sequencing.

### 5.5. Recombinant SVMP Expression and Purification

Ecarin and EoMP06 sequences obtained from Uniprot (Q90495 and Q6X1T6, respectively) were codon optimized for mammalian expression, and cloned into a variant of pcDNA3.4 containing an N-terminal signal sequence (V_H_1-2 leader) and a C-terminal His_10_-tag and Avitag by Genscript (Piscataway, NJ, USA). rEcarin and rEoMP06 (with the complete prodomain) were expressed in HEK293 cells (Expi293, Thermo Fisher Scientific, Waltham, MA, USA) at a 100 mL scale using FectoPRO (Polyplus Transfection, Illkirch, France). Cells were grown for 5 days with shaking at 240 rpm, 37 °C and 8% CO_2_ in Expi293 expression medium (Thermo Fisher Scientific, Waltham, MA, USA). Approximately 24 h after transfection, cells were fed 0.4% D-(+)- glucose solution and 3 mM sodium valproic acid solution. Five days post-transfection, the supernatant was harvested from cell cultures, filtered through a 0.45 μm filter and incubated for 30 min, rotating at 4 °C with HisPur nickel–nitrilotriacetic acid (Ni-NTA) resin (Thermo Fisher Scientific, Waltham, MA, USA). Recombinant SVMPs were eluted with 250 mM imidazole and dialyzed against PBS (or Tris pH8 50 mM, NaCl 50 mM for cryo-EM experiment) at 4 °C. Constructs were then aliquoted, frozen in liquid nitrogen and stored at −80 °C. A 100 mL culture typically yields 0.3–0.6 mg of pure (>90%) rSVMP.

To obtain biotinylated SVMPs, the same protocol was followed but the plasmid DNA of the SVMPs was co-transfected with a bifunctional biotin-[acetylCoA carboxylase] holoenzyme synthetase/DNA-binding transcriptional repressor, bio-5′-AMP-binding (BirA) expression plasmid into Expi293 cells (Thermo Fisher Scientific, Waltham, MA, USA, cat #A14527). In addition to VPA and glucose, cells were also supplemented with 20 nM d-biotin 24 h after transfection.

### 5.6. SVMP Rate Calculation

Various concentrations of PT (depending on the assay) were mixed with 0.5 mM of S-2238 substrate in PBS (pH 7.4). The reactions were started by adding the SVMP at 37 °C, and the absorbance at 405 nm (Abs405), which corresponds to pNa formation, was monitored. The rates were calculated from the slope of the first derivative, d(pNA)/dt versus time. The reactions were monitored at 3 s intervals, allowing the calculation of the second derivative, which represents the reaction rate of ecarin [51]. We used GraphPad (Prism) for the calculations, using the Savisky–Golay smoothing factor with the 4th polynomial order and a window width of nine data points.

### 5.7. Antibody Expression and Purification

We reformatted 96 Fabs selected by NGS as IgG1, and we synthesized those by GenScript (Piscataway, NJ, USA) into a mammalian expression vector containing both the HC and LC separated by a P2A self-cleaving motif. Similarly to rSVMPs, antibodies were expressed in HEK293 cells (Expi293, Thermo Fisher Scientific, Waltham, MA, USA). The supernatants of 24-deep well culture (4 mL) supernatants were harvested after five days and purified using protein A magnetic beads (Thermo Fisher Scientific, Waltham, MA, USA). Antibodies were tested for both binding to ecarin or polyspecific preparations and neutralization of ecarin and EoMP06. Selected antibodies (B3, H11 and H12) were re-expressed at a medium scale (30 mL) and IgG-purified on Protein A Sepharose (Cytiva, Marlborough, MA, USA). Constructs were dialyzed against PBS and stored at 4 °C.

### 5.8. Recombinant Ecarin ELISA

ELISA with His_10_-tagged recombinant ecarin as an antigen was performed according to the protocol described in [55]. Absorption was measured at 405 nm after less than 15 min. Positive and negative controls were systematically used. The absorbance values for the highest antibody concentration (0.1 mg/mL) are depicted in Figure 1.

### 5.9. Polyspecificity Reagent ELISA

According to the protocol described in [78], both solubilized CHO-cell membrane proteins (SMPs) and single-strand (SS) DNA (Sigma-Aldrich, St. Louis, MO, USA, cat # D8899) were used to assess the polyspecificity of our antibodies in PSR ELISA. Absorption was measured at 405 nm after 15 min. The absorbance values for the highest antibody concentration (0.1 mg/mL) are depicted in Figure 1.

### 5.10. Epitope Binning by BLI

Binding assays were performed using an Octet HTX system (Sartorius, Goettingen, Germany). Ligand immobilization, binding reactions and washes were conducted in wells of black polypropylene 96-well microplates (Greiner, Monroe, NC, USA, cat # 655209). We captured 50 nM of His_10_-tagged rEcarin using anti-Penta-HIS biosensors (Fortébio, Menlo Park, CA, USA, cat # 18-0038). After rEcarin loading for 5 min (to produce a signal of ~1 nm), a saturating concentration (0.1 mg/mL) of monoclonal antibodies (B3, H11 or H12) was added for 3 min. Competing concentrations of the same three antibodies (0.05/mL) were then added for 3 min to measure binding. All incubation steps were performed in PBS with 0.1% TWEEN 20 and 0.1% BSA. Binding (increase in signal) of the competing antibody indicates an available epitope, while no binding (no change in signal) indicates a blocked epitope.

### 5.11. Binding Kinetics by BLI

Binding assays were performed using an Octet HTX system (Sartorius). Ligand immobilization, binding reactions and washes were conducted in wells of black polypropylene 96-well microplates (Greiner 655209). For conventional kinetic/dose–response tests, IgG1s (B3, H11 and H12) were immobilized on AHC2 sensors (Sartorius, Goettingen, Germany, cat # 18-5142) to ~1 nm (30 s). Binding analysis of a concentration series of rEcarin (31–1000 nM) was carried out at 30 °C and 1000 rpm, with a 180 s association followed by a 180 s dissociation. All incubation steps were performed in PBS with 0.1% TWEEN 20 and 0.1% BSA. Data were analyzed using Analysis Studio software Version 12.2 (Sartorius) with Savitzky–Golay filtering. A partial fitting where the association and dissociation steps were fitted separately was used to calculate the *K*_D_ values of the three antibodies. A 1:1 model was used (Appendix A).

### 5.12. H11 Fab Preparation for Cryo-EM

To generate functional H11 Fab, a full H11 IgG1 construct was digested with papain (Sigma-Aldrich, St. Louis MO, USA, cat # P3125) at a 1:75 papain-to-IgG1 mass ratio for 2.5 h at 37 °C. The reaction was quenched with iodoacetamide (Sigma-Aldrich, St. Louis, MO, USA, cat # 407710) and dialyzed against 50 mL Tris pH 8.0, NaCl 50 mM. Lastly, the non-digested IgG1 and cleaved Fc were removed by protein A agarose beads (Cytiva, Marlborough, MA, USA). The purity of H11 Fab was confirmed by SDS-PAGE (Appendix A).

### 5.13. Venoms

Snake venom was sourced from animals maintained in the herpetarium of the Liverpool School of Tropical Medicine (LSTM). This facility and its snake husbandry protocols are approved and inspected by the UK Home Office (establishment license number 40/9074 [X2OA6D134]) and LSTM’s Animal Welfare and Ethical Review Body. Venom was extracted from three snake species held in the facility, two African, namely, the vipers *E. p. leakeyi* (Kenya) and *E. romani* (Nigeria), and one Indian, *E. carinatus sochureki* (referred to throughout as *E. carinatus*). Venom from *E. leucogaster* (Mali) was sourced from the historical venom collection held at the LSTM. All venoms were from snakes of adult size and included both sexes, except for *E. carinatus*, where the venom was from a single specimen that was inadvertently imported to the UK via a boat shipment of stone, and then rehoused at the LSTM on the request of the UK Royal Society for the Prevention of Cruelty to Animals (RSPCA). Venoms were lyophilized and stored at 4 °C until resuspension in PBS for downstream experiments.

### 5.14. Stock Preparation of Commercially Available Purified Ecarin

We resuspended 0.4 mg of ecarin (Sigma E0504) in 400 µL PBS (stock concentration of 1 mg/mL). Moreover, we used 0.1 µg of commercial ecarin in our PT activation assay as this quantity demonstrated similar activity to 2 nM rEcarin. This would therefore enable a meaningful comparison between commercial and recombinant ecarin.

### 5.15. Native SVMPIII Isolation from Crude Echis Venoms by SEC

We resuspended 4–6 mg of lyophilized *Echis* venoms in 0.4–0.6 mL of PBS pH 7.4 (final concentration 10 mg/mL) at 4 °C. The insoluble proteins were removed by centrifugation (10 min, 21,000 rpm), and the supernatant was applied to a Superdex S200 Increase 10/300 GL column (Cytiva, Marlborough, MA, USA) pre-equilibrated in the resuspension buffer. Elution was carried out with the same buffer at 4 °C. The flow rate was 0.5 mL/min. The fractions with distinct molecular weights above 50 kDa were combined. The concentrations of the native SVMP PIIIs for the PT activation assay were estimated using the molecular weight and extinction coefficient calculated from the ecarin sequence (51.5 kDa and 60,915 M^−1^·cm^−1^).

### 5.16. Sample Preparation for Cryo-EM

For all samples, ecarin and H11 were diluted to 2 μΜ (~0.1 mg/mL) and mixed at a 1:1 ratio, resulting in 2 μΜ of complex (approximately 0.2 mg/mL). Samples prepared for 300 mesh R 1.2/1.3 UltrAuFoil Holey Gold Films (Quantifoil, Jena, Germany) were concentrated to a final concentration of 20 μΜ (~2 mg/mL) before vitrification, while 2 μΜ of complex was used for graphene-coated 300 mesh R 0.6/1 UltrAuFoil Holey Gold Films (Quantifoil, Jena, Germany), which were prepared as previously described [56]. Four microliters of the sample were applied to films before vitrification using a Vitrobot Mark IV system (blot time 6 s (holey gold) or 5 s (graphene), blot force 2, 100% humidity, room temperature, Whatman No. 1 filter paper). The 300 mesh R1.2/1.3 UltrAuFoil Holey Gold Films (Quantifoil, Jena, Germany)) were glow discharged for 25 s at 15 mA with a Pelco Easiglow 91,000 (Ted Pella, Inc., Redding, CA, USA) in an ambient vacuum. Graphene grids were made hydrophilic with UV/ozone treatment using the UVOCS T10×10 system. We first performed a 10 min ”warmup” run before inserting and treating grids for 4 min.

### 5.17. Cryo-EM Data Acquisition

Data were collected on a Thermo-Fisher Talos Arctica transmission electron microscope operating at 200 keV equipped with a Gatan K2 Summit direct electron detector after setting parallel illumination conditions [79]. We acquired 5 s exposures using an exposure rate of 3.46 e^-^/pixel/sec (10.8 e^-^/Å2/s) divided into 25 frames in the counting mode, resulting in a total electron exposure of 54 e^-^/Å2. The Leginon data collection software [80] was used to collect micrographs at 73,000× nominal magnification (0.566 Å/pixel at the specimen level) with a nominal defocus range of 0.9 μm to 1.6 μm. Stage movement was used to target the center of 16 holes (holey gold) or 49 holes (graphene grid), and coma-compensated image-beam shift [81] was used to acquire high-magnification images in the center of each hole. A total of 2893 micrographs were collected over holey gold while a total of 2618 (graphene) micrographs were collected over graphene. Micrograph frames were aligned using MotionCor2 [82] and processed in real-time using CryoSPARC live [83] to monitor image quality during data acquisition. See Appendix A for details.

### 5.18. Image Processing

The general processing workflow is described here, and further details are provided in the processing workflows in Appendix A. For both datasets, movie frames were motion-corrected and dose-weighted using MotionCor2 [82], and the summed and dose-weighted micrographs were imported into cryoSPARC v4.3 [83] for patch CTF correction. Micrographs with estimated defocus values greater than 3.0 μm or with estimated resolutions worse than 7 Å resolution were discarded. Initial rounds of picking were performed using a blob picker with an upper radius of 200 and lower radius of 50. Particles were extracted using a 540-pixel size box. We used 2D classification to remove particle selections contributing to obviously bad classes, and the resulting stack of particles was used to generate three ab initio models using default parameters. The cleaned particle stack was then sorted using heterogeneous refinement with one good 3D class and two “trash” 3D classes.

For the graphene dataset, the blob picker (upper radius of 200 and lower radius of 50) was used to generate an initial stack, which was then lightly cleaned using 2D classification with standard settings. The resulting particles were then sorted using heterogenous refinement, using one good 3D class and two “trash” 3D classes. The particles from the good class were then combined with the good particles from the holey gold dataset for another round of heterogeneous refinement with four good 3D classes. The particle stack with the highest resolution and best angular distribution was then reextracted and subjected to a round of 2D classification using standard settings, except for a batch size per class of 400 and 40 online OEM iterations. The resulting classes were improved, and 8 templates were selected for template picking using a particle diameter of 90 angstroms, an NCC cutoff of 0.0–0.3 and a power score cutoff of 165–350. The resulting 1.8 million particle stack was extracted into a 540-pixel size box and subjected to a round of heterogeneous refinement with 1 good 3D class and 4 trash 3D classes to remove bad picks and damaged particles. The resulting 517K particles from the good heterogeneous refinement class were then combined with 298K particles from the good heterogeneous refinement class of the holey gold dataset and subjected to another round of heterogenous refinement with two good classes and three trash classes. The good class with the highest resolution and best particle distribution was then refined using the NU-refinement job with optimize per-particle defocus and optimize per-group CTF parameters (tilt and trefoil) activated [84]. A mask containing ecarin and the heavy- and light-chain variable regions of the Fab was then used in a local refinement job (rotation search extent 1°, shift search extent 1 Å, maximum alignment resolution of 0.1° and initial lowpass filter of 5 Å) to generate the final consensus volume.

### 5.19. Atomic Model Building and Refinement

Starting coordinates for the atomic model of ecarin–H11 were generated by manually docking the structure of a V_H_1-2 class antibody (PDB 4Q2Z) and an AlphaFold 2.0 [85]-generated model of ecarin into the available EM density using ChimeraX-1.8 [86]. Iterative rounds of model building and refinement were performed in PHENIX v1.19.2 [87] and Coot 0.9.4.1EL [88] until reasonable agreement between the model and data was achieved. Final model relaxation and removal of clashes and bad rotamer outliers were performed using ISOLDE [89], followed by one more round of refinement in PHENIX. ChimeraX [86] was used to interpret EM maps and models, as well as to generate figures.

## Figures and Tables

**Figure 1 toxins-16-00361-f001:**
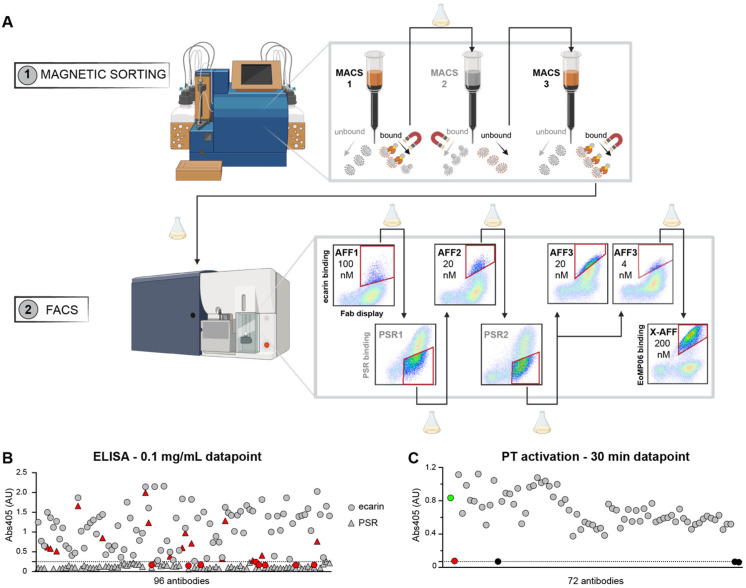
Isolation of rEcarin neutralizing antibodies. (**A**) Sorting strategy for the selection of high-affinity rEcarin binders and cross-rEcarin/rEoMP06 binders. rEcarin was used as bait in MACS. For FACS, iterations of positive selections with rEcarin and depletions with polyspecific reagent (PSR) were used to remove nonspecific antibodies. The final AFF sort (X-AFF) was performed with rEoMP06 to select for enriched cross-reactive clones. Created with BioRender.com. (**B**) Selection of antibodies for functional assay by ELISA. The highest tested concentration (0.1 mg/mL, 0. 67 µM) is shown for each construct. Antibodies highlighted in red color were not chosen either because they did not bind to rEcarin (circles) or bound to PSR preparations (triangles). A threshold is set at Abs405 = 0.25 AU. (**C**) Identification of three rEcarin neutralizing antibodies. Reaction conditions: PBS pH 7.4, 37 °C, rEcarin 2 nM, recombinant human PT 0.2 µM, S-2238 0.5 mM, antibody 0.1 mg/mL (0.67 µM). The Abs405 after 30 min reaction time is depicted. Positive (green dot) and negative (red dot) controls were performed in the absence of antibody and in the presence of 20 mM EDTA, respectively. Three antibodies (B3, H11 and H12, black dots) show a similar inhibition to 20 mM EDTA (dotted line).

**Figure 2 toxins-16-00361-f002:**
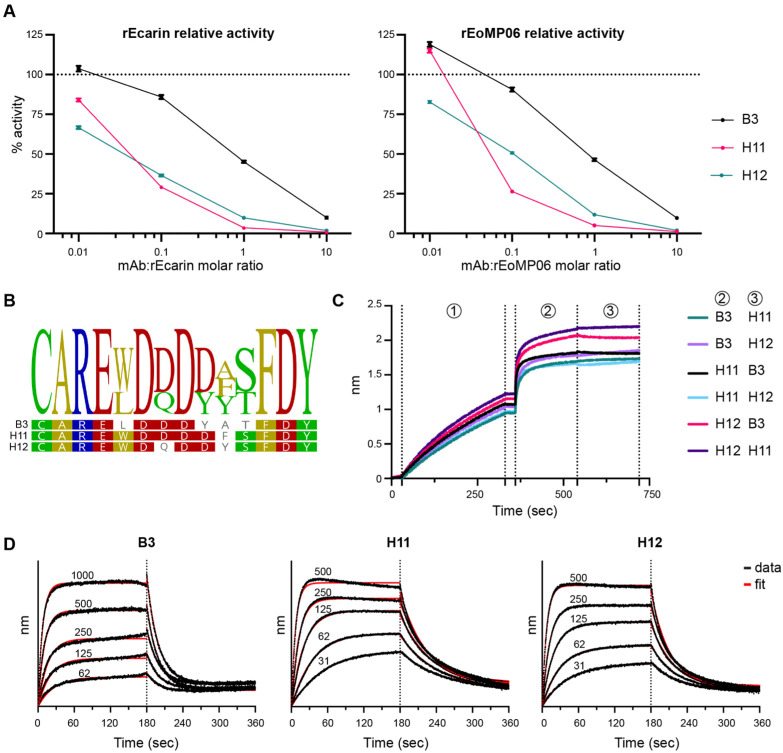
Characterization of rEcarin and rEoMP06 neutralizing antibodies. (**A**) Relative activity of rSVMP (rEcarin, left, and rEoMP06, right) in the presence of different concentrations of B3, H11 and H12. Reaction conditions: PBS pH 7.4, 37 °C, rEcarin 50 nM (**left**) or rEoMP06 50 nM (**right**), recombinant human PT 0.2 µM, S-2238 0.5 mM, antibody 0.5–500 nM. We set 100% activity as the reference rate for each SVMP in the absence of antibody. Reactions were run in biological triplicate. The errors bars represent the standard deviations of the measurements. (**B**) CDRH3 sequence alignment of B3, H11 and H12. (**C**) Epitope binning showing competition between B3, H11 and H12 for ecarin binding. Step 1: rEcarin immobilized on the Penta-His sensor, step 2: saturating antibodies (0.1 mg/mL) bind to ecarin, and step 3: competing antibodies (0.05 mg/mL) bind to ecarin to assess competition (see Section 5.10 for details). Six different combinations of antibodies are depicted in distinct colors across the three steps. None of the competing antibodies (step 3) bind to ecarin, which indicates that they recognize the same or overlapping epitope as the saturating antibodies (step 2). (**D**) BLI sensorgrams depicting rEcarin binding to B3, H11 and H12. The concentrations of rEcarin are indicated on the traces (31–1000 nM). Raw and fitted data are shown in black and red, respectively.

**Figure 3 toxins-16-00361-f003:**
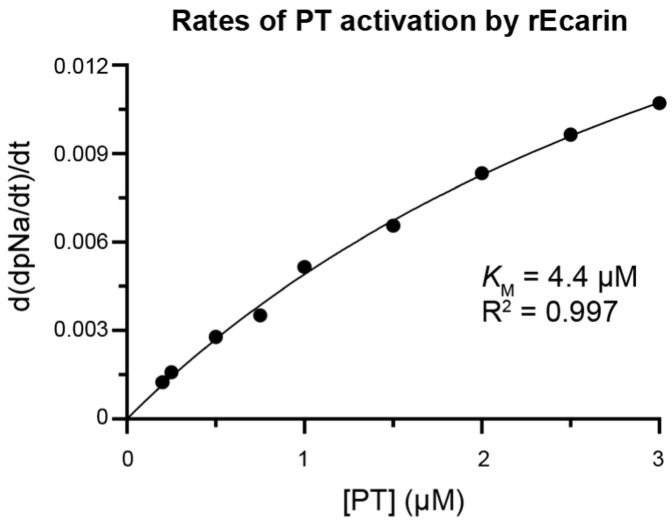
Michaelis–Menten curve of PT activation catalyzed by rEcarin. Reaction conditions: PBS pH 7.4, 37 °C, rEcarin 50 nM, recombinant human PT 0.2–3 µM, S-2238 0.5 mM. Reactions were carried out in duplicate for each data point.

**Figure 4 toxins-16-00361-f004:**
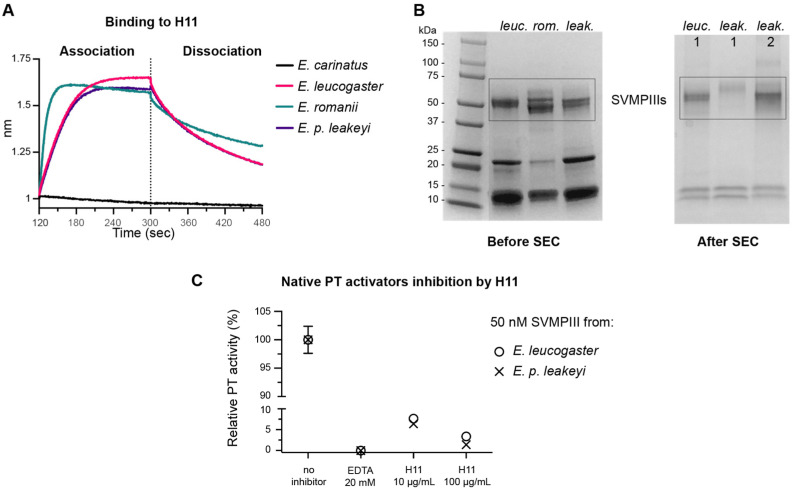
H11 inhibits nEcarins from African saw scale vipers. (**A**) H11 binds to whole venoms of *E. leucogaster* (Mali), *E. p. leakeyi* (Kenya) and *E. romani* (Nigeria), but not *E. carinatus* (India). (**B**) SDS PAGE gels in reducing conditions showing the isolation of SVMP PIIIs from *E. leucogaster* and *E. p. leakeyi* after SEC. (**C**) Inhibition of nEcarins from *E. leucogaster* and *E. p. leakeyi* venoms. Reaction conditions: PBS pH 7.4, 37 °C, isolated SVMP PIII fractions 50 nM, recombinant human PT 0.2 µM, S-2238 0.5 mM, inhibitor concentrations are indicated on the graph. We set 100% activity as the reference rate for each SVMP in the absence of inhibitor. Reactions were performed in duplicate. The error bars indicate the standard deviations of the measurements.

**Figure 5 toxins-16-00361-f005:**
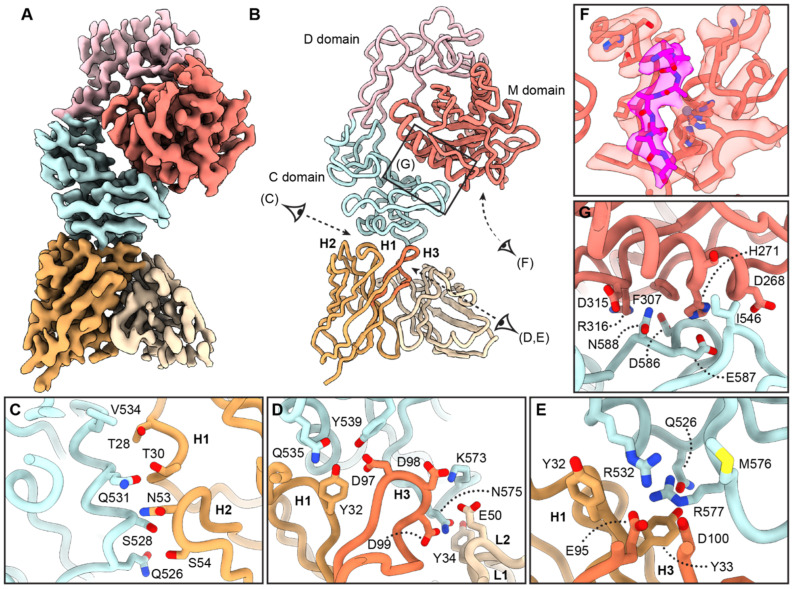
Cryo-EM structure of H11 Fab bound to ecarin. (**A**) Overview of the sharpened cryo-EM map of H11 Fab–ecarin complex. The map is segmented and colored according to domain architecture (M domain, salmon; D domain, pink; C domain, pale cyan; H11 Fab V_H_, orange; H11 Fab V_L_, light orange). (**B**) Overview of the atomic model generated using the cryo-EM density. The coloring scheme matches the same domain colors in panel A except that the CDRH3 loop, which contains the diversity in our antibody library, is colored coral. Perspective eyes denote the viewing angle for panels (**C**–**F**) and the black box provides the viewing perspective for panel (**G**). (**C**–**E**) Interface interactions between H11 and the C domain of ecarin. (**F**) Modeled polyalanine peptide in the M domain proteolytic active site with associated cryo-EM density at a binarization threshold of 0.2. The peptide is rendered as magenta, and the M domain is salmon. (**G**) Interactions between the C domain and M domain of ecarin corresponding to the black box in panel B. All interacting residues were selected using a distance cutoff of 4 Å between the two domains.

**Figure 6 toxins-16-00361-f006:**
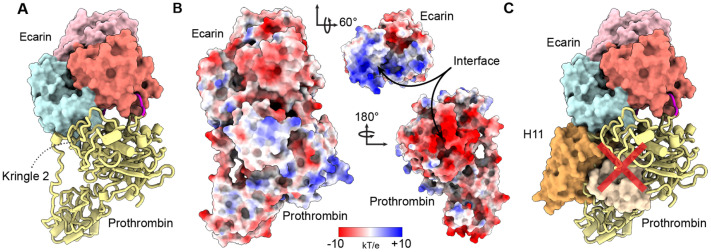
AlphaFold 2 complex of PT–ecarin. (**A**) Overview of the highest-scored ecarin–PT complex from AlphaFold 2. Ecarin is represented as a surface and rendered with the M domain as salmon, D domain as pink and C domain as pale cyan. The Kringle 2 domain of PT (yellow) associates with the C domain (pale cyan) of ecarin in our predicted model. The loop containing the secondary ecarin cut site bound in the M domain active site is colored magenta. (**B**) Electrostatic potential surfaces of the predicted ecarin–PT AlphaFold model. Ecarin and PT present complementary charged surfaces to facilitate complex formation. (**C**) The predicted PT–ecarin model overlaid with our H11 Fab–ecarin cryo-EM structure, demonstrating significant clashes (red X) between H11 (V_H_ orange and V_L_ pale orange, surface representation) and PT (yellow, cartoon representation) at the C domain interface.

## Data Availability

Data that support the findings of this study can be made available from the corresponding author upon request. Cryo-EM maps were deposited to the Electron Microscopy Data Bank (EMDB) under accession code EMD-45728. Atomic coordinates for the focused refinement model were deposited to the PDB under accession code 9CLP. The plasmids described in the manuscript will be made available by MTA.

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
