# Peer review of "Importance of the Cysteine-Rich Domain of Snake Venom Prothrombin Activators: Insights Gained from Synthetic Neutralizing Antibodies"

_toxins, 2024, doi:10.3390/toxins16080361_

Round 1

Reviewer 1 Report

Comments and Suggestions for Authors

The manuscript titled “Importance of the cysteine-rich domain of snake venom prothrombin activators: Insights gained from synthetic neutralizing antibodies” (toxins-3127782) is based on numerous experimental data and might provide useful information on the mode of action and/or approaches to inactivate specific snake venom metalloproteinases (SVMPs) that exert their poisonous function through prothrombin activation. In my opinion, a weak point of the manuscript is the Discussion section, which should avoid repeating literature information (e.g., in the first paragraph) and be more focused on how to further explain/elucidate/comment on the various experimental results presented. Overall, the manuscript may eventually be published in Toxins -provided that a series of issues will be successfully addressed by the Authors.

Major Comment

Discussion should shed light and/or critically comment on a series of original results included in the manuscript. In my opinion, a specific paragraph summarizing and discussing/ explaining/ addressing the unexpected results obtained -and presented here and there in the text- as well as clearly mentioning any limitations of the whole study would be very useful and should be added to the Discussion part (please, see also points 8 and 9, in “Other Comments”).

Other Comments

1. p. 2, line 97, “Structural analysis of one neutralizing antibody with ecarin…”: please, add the origin/source of ecarin used.

2. p. 4, Figure 1B, Y-axis, “Abs450 (AU)”: is 450 correct? Please, check.

3. p. 4, line 167, “A threshold is set at Abs450…”:  is 450 correct? Please, check.

4. p. 4, lines 176-177, “The assay was set up as described above but with the addition of a single high concentration (0.1 mg/mL) of antibody”: Please, describe the assay set up in detail and/or refer to the relevant paragraph in Materials and Methods.

5. p. 5, lines 189-190, ”2: binding to ecarin of 0.1 mg/mL saturating antibodies and, and 3: binding to ecarin of 0.05 mg/mL competing antibodies.”: Please, define the terms “saturating” and “competing” antibodies (in both cases, B3, H11, and H12) to facilitate understanding, and/or refer to paragraph 5.1 in Materials and Methods.

6. p. 6, lines 216-217, “The error bars indicate the standard deviations on the measurements”: No error bars can be observed in Figure 3 -am I wrong?

7. p. 6, line 218, “H11 neutralizes native ecarin (nEcarin) from African Echis venoms”: It might be better to change into: H11 neutralizes native ecarin (nEcarin) in-house purified from African Echis venoms

8. p. 9, lines 331-333, “Prodomain peptides could potentially be present in our ecarin preparation [39, 48, 49] and might have been inadvertently trapped in our ecarin-H11 complex”: In my opinion, it would be helpful if you could further discuss this issue, e.g. in the Discussion section.

9. p. 11, lines 452-454, “In conclusion, our study highlights the utility of combining recombinant and native PT activators to identify neutralizing epitopes, emphasizing the relevance of non-catalytic domains as crucial targets”: It would be helpful if you could further explain (preferably in the Discussion part) how exactly the combined use of native and recombinant PT activators may provide complementary results on the identification of neutralizing epitopes.

10. p. 14, line 591, “was measured at 450 nm”: please, confirm; is it 450nm or 405 nm?

11. p. 15, line 626, “H11 Fab/ecarin complex preparation for cryo-EM”: It might be better to change into: H11 Fab preparation for cryo-EM samples

12. p. 15, line 647, “Purified ecarin stock preparation”: It may be better to change into: Stock preparation of commercially available purified ecarin 

13. Figure S1, B: This is difficult to follow; legend should provide more information to the reader. What is the difference between schemes shown in the 2nd and 4th row?

Comments on the Quality of English Language

Minor editing of English language is required.

Reviewer 2 Report

Comments and Suggestions for Authors

Two minor comments: 

1. Most people are not expert in venom toxin classifications, but as soon as they read "Cysteine Rich" will think "CRISP" and then wonder the relationship between CRISP and svMPs in this paper. Suggest paying homage to the idea that in a young field, there should be something to educate everyone even in the most specialized paper.  

2. Figure 4 legend makes no mention of E. carinatus, but shows E. carinatus in the figure. Figure should stand alone in visually and by explanation. The reader should not be compelled to go back to the text to figure out why it's not discussed in the legend or if it is simply an omission. I THINK I know why it's not discussed further in the figure, but it would be a lighter read and more educational if the authors would simply explain, "this is different because"...I think it's most useful to have it in the figure, so my preference as a reader would simply be to have a better understanding about why it's different and why I shouldn't be concerned (or why I should/limitation etc). 

Round 2

Reviewer 1 Report

Comments and Suggestions for Authors

The authors’ point-to-point response is clear and persuasive. All issues have been addressed and “confusing points” of the original version (e.g. some schemes shown in Figure S1.B, Abs measured at 450 or 405 nm) have been corrected. The revised manuscript is clearer and can better bring the authors’ tedious work into the readers’ view.

The "minor language editing" initially suggested referred to the correction of some typing errors [e.g., bacto-cassamino (line 469), SVMPIII (line 638 and elsewhere), Holy (line 652), either ml or mL, either hour or h, etc., throughout the text]. Otherwise, the English language used is fine.